# Long- and Short-Term Trends in Outpatient Attendance by Speciality in Japan: A Joinpoint Regression Analysis in the Context of the COVID-19 Pandemic

**DOI:** 10.3390/ijerph20237133

**Published:** 2023-12-01

**Authors:** Asuka Takeda, Yuichi Ando, Jun Tomio

**Affiliations:** 1Department of Health Crisis Management, National Institute of Public Health, Wako-shi, Saitama 3510197, Japan; 2Department of Health Promotion, National Institute of Public Health, Wako-shi, Saitama 3510197, Japan

**Keywords:** outpatient attendance, speciality, clinic, COVID-19, pandemic, joinpoint regression

## Abstract

The COVID-19 pandemic resulted in a decline in outpatient attendance. Therefore, this study aimed to clarify long- and short-term clinic attendance trends by speciality in Japan between 2009 and 2021. A retrospective observational study of Japan’s claims between 2009 and 2021 was conducted using the Estimated Medical Expenses Database. The number of monthly outpatient claims in clinics was used as a proxy indicator for monthly outpatient attendance, and specialities were categorised into internal medicine, paediatrics, surgery, orthopaedics, dermatology, obstetrics and gynaecology, ophthalmology, otolaryngology, and dentistry. The annually summarised age-standardised proportions and the percentage of change were calculated. Joinpoint regression analysis was used to evaluate long-term secular trends. The data set included 4,975,464,894 outpatient claims. A long-term statistically significant decrease was observed in outpatient attendance in internal medicine, paediatrics, surgery, ophthalmology, and otolaryngology during the pandemic. From March 2020 to December 2021, which includes the COVID-19 pandemic period, outpatient attendance in paediatrics, surgery, and otolaryngology decreased in all months compared with that of the corresponding months in 2019. For some specialities, the impact of the pandemic was substantial, even in the context of long-term trends. Speciality-specific preparedness is required to ensure essential outpatient services in future public health emergencies.

## 1. Introduction

The World Health Organization (WHO) declared the COVID-19 pandemic on 11 March 2020 [1]. Countries needed to minimise the disruption of healthcare services and achieve an optimal balance with the COVID-19 response [2]. During the early stages of the pandemic, outpatient attendance decreased substantially worldwide, especially among vulnerable populations such as children and older adults [3,4,5]. Certain demographic groups, such as women, urban residents, non-smokers, non-drinkers, and those without chronic illnesses, were more likely to delay health check-ups and non-urgent healthcare services during the pandemic [6]. The utilisation of healthcare services during this time varied depending on the type of medical insurance, with financial security becoming a crucial factor [7].

The WHO has set operational guidelines for maintaining essential health services during an outbreak [8]. According to these guidelines, countries should set indicators and monitor them at regular intervals in a timely manner to ensure close monitoring of essential health services. One sample indicator is the number of outpatient attendances. The WHO conducted three pulse surveys on essential health services during the COVID-19 pandemic until 2021. The first pulse survey, conducted between May and July 2020, recorded reductions in outpatient attendance due to lower demand in 76% of all countries [9]. The second (January–March 2021) and third (November–December 2021) reported that nearly all countries continued to be affected by the COVID-19 pandemic in terms of essential health services, including disruption of services [10,11].

The COVID-19 pandemic considerably impacted healthcare systems and the delivery of healthcare services. The literature reports short-term trends by speciality separately in the early stages of the pandemic [12,13,14,15,16]. Most studies also reported that outpatient attendance has recovered substantially as of 2021. However, the WHO reported that the pandemic led to continued service interruptions until the end of 2021. This was possibly due to the high volume of studies on secondary care and the limited number of studies on the use of general medical services and access to primary care [17]. Observing outpatient attendances over time, such as over the course of a decade, including the period of the COVID-19 pandemic, may help clarify the impact of the pandemic in the context of long-term trends. In addition, trends in outpatient attendance by speciality have been reported separately, not collectively, and few studies have compared two or more medical specialties [18]. However, in an outpatient system like that in Japan, where there is no general practice for primary care and clinics with different specialities provide outpatient care, the impact of the pandemic on outpatient attendance by speciality must be understood for future preparedness.

Examining the impact of COVID-19 on outpatient attendance requires an investigation of long-term trends including the pre-pandemic period. Japan has a strong universal health insurance programme, and the claims data are large and include all citizens. These claims data can be used to identify not only the number of claims but also the cost per day and the days per claim and to determine differences in trends by speciality.

This study aimed to examine the long-term trends in outpatient attendance by speciality in Japanese clinics between 2009 and 2021 and the short-term trends between 2020 and 2021. The study also investigated changes in trends during the COVID-19 pandemic.

## 2. Materials and Methods

### 2.1. Study Design

We conducted a retrospective observational study, as we aimed to investigate the past trends of outpatient attendance and the impact of the COVID-19 pandemic using the Estimated Medical Expenses Database (EMED), the administratively collected national-level claims database.

### 2.2. Data Source

Open data from EMED provided by Japan’s Ministry of Health, Labour, and Welfare (MHLW) were used. Japan has an established universal health insurance system, and the EMED includes nationwide data. The EMED started as of April 2000 to quickly grasp trends in national medical expenses and currently tabulates medical expenses for medical inpatients and outpatients; dental, pharmacy, hospital meal and living expenses; and home-visit nursing expenses [19]. The EMED is a monthly report based on claims compiled before review by all insurers. We used aggregate open data between 2009 and 2021 comprising the number of outpatient claims at clinics by speciality. The aggregate open data also included the total number of claims, the total cost, and the total number of days. Specialities were categorised into internal medicine, paediatrics, surgery, orthopaedics, dermatology, obstetrics and gynaecology, ophthalmology, otolaryngology, dentistry, and others. If a clinic had multiple specialities, all visits were classified into one ‘main’ speciality. Three age groups were extracted: children 0–5 years old, people 6–69 years old, and people 70 years and older. This study excluded data that were publicly funded or classified under other clinical specialities.

Moreover, to obtain the baseline population, we used population estimates from the Statistics Bureau of Japan that were retrieved on 1 October of each year.

### 2.3. Variables for Outpatient Attendance by Speciality

The number of monthly outpatient claims was used as a proxy for monthly outpatient attendance. In addition, the cost per day and the days per claim (based on monthly values) were used as supplementary variables that reflected the status of outpatient care.

### 2.4. Statistical Analysis

The annually summarised age-standardised proportions for joinpoint analysis between 2009 and 2021 were calculated as the average age-standardised proportion each month. Age-standardised proportion was defined as the number of outpatient claims per population by speciality using an age-class-specific model population of Japan in 2015 [20]. To classify the long-term trend in outpatient attendance, the age-standardised proportion and the standard errors were fitted by speciality using joinpoint regression analysis. Joinpoint trend analysis software, developed at the National Cancer Institute in the United States, was used for conducting a temporal trend analysis [21]. Joinpoint regression analysis comprises continuous linear segments that are tested to determine whether an apparent change in the trend is statistically significant. The significance tests were conducted using the Monte Carlo Permutation method, in which the models potentially incorporated the estimated variability for each data point [22]. The trend change points (joinpoints), annual percent change (APC), and average annual percent change (AAPC) were calculated. After determining the number of joinpoints, the various models with k joinpoints are evaluated by calculating their Bayesian Information Criterion [23]. If there are no joinpoints, indicating no shifts in trend, APC remains constant and is equal to the AAPC. Alternatively, when there are trend changes, the entire period is divided into segments based on these points. Subsequently, the AAPC is calculated as a weighted average of the estimated APC within each segment, with segment lengths serving as the weights [24]. Statistical significance was set at *p* < 0.05. Percent changes were calculated as the number of claims in a month during a state of emergency minus that in the corresponding month in 2019, divided by that in the corresponding month in 2019 and multiplied by 100. In Japan, an emergency was declared three times: in April–May 2020, January–March 2021, and April–September 2021.

The percentage of changes in the number of monthly claims, cost per day, and days per claim between 2020 and 2021, and the corresponding periods in 2019 by speciality were calculated. We compared the percentage of change in the number of outpatient claims by speciality for 2020–2021 to determine the months that saw the greatest decrease in claims. Moreover, we compared the percentage of changes in the number of outpatient claims, cost per day, and number of days per claim for each speciality.

Analyses were performed using Stata/MP Version 16.1 (StataCorp, College Station, TX, USA) and Joinpoint Regression Program Version 4.9.1.0 (National Cancer Institute, Bethesda, MD, USA).

This study did not require an institutional review board approval because it is an observational study that used only aggregate public domain data.

## 3. Results

The data set included 4,975,464,894 outpatient claims between 2009 and 2021. The summary of the annual number of outpatient claims by speciality is shown in Appendix A. The results of the joinpoint regression analysis are shown in Table 1. Joinpoint regression results were classified into four types according to trend characteristics: (1) increased/stabilised in the pre-pandemic period and decreased during the pandemic; (2) decreased since the pre-pandemic period; (3) increased in the pre-pandemic period and stabilised during the pandemic; and (4) increased since the pre-pandemic period (Figure 1). For each pattern, a joinpoint analysis was used to identify statistical significance by speciality. Type 1 included internal medicine, paediatrics, ophthalmology, otorhinolaryngology, and dentistry. Type 2 included surgery. Type 3 included orthopaedics and dermatology. Type 4 included obstetrics and gynaecology. Internal medicine (2018–2021 APC: −2.86; 95% confidence interval (CI): −4.78, −1.51), paediatrics (2019–2021 APC: −6.91; 95% CI: −12.93, −0.85), surgery (2019–2021 APC: −9.53; 95% CI: −10.65, −8.60), ophthalmology (2018–2021 APC: −2.72; 95% CI: −4.99, −1.37), and otolaryngology (2019–2021 APC: −9.42; 95% CI −14.72, −3.34) showed a statistically significant decrease in the number of outpatient claims (the age-standardised proportion) during the pandemic. The specialities with statistically significant APCs in the joinpoint regression analysis between 2009 and 2021 were internal medicine, paediatrics, surgery, obstetrics and gynaecology, and otolaryngology. The specialities in which AAPC showed a statistically significant increase in the number of outpatient claims were orthopaedics (AAPC: 1.40; 95% CI: 1.06, 1.68), dermatology (AAPC: 1.95; 95% CI; 1.81, 2.12), obstetrics and gynaecology (AAPC: 1.20; 95% CI: 0.92, 1.34), and dentistry (AAPC: 1.92; 95% CI: 1.58, 2.16). In contrast, the departments in which the AAPC showed a statistically significant and decreasing trend in the number of recipients were internal medicine (AAPC: −0.36; 95% CI: −0.61, −0.13), surgery (AAPC: −3.60; 95% CI: −3.78, −3.47), and ophthalmology (AAPC: −0.75; 95% Cl: −1.07, −0.48).

The number of outpatient claims per 1,000 people began to decline in all specialities in March 2020, shortly after the COVID-19 pandemic was declared (Table 2). For all specialities, the greatest decrease in outpatient visits during the pandemic occurred in April 2020 or January 2021. The greatest decreases in the number of monthly claims compared with those in the corresponding month in 2019 occurred in April 2020 for orthopaedics, dermatology, obstetrics and gynaecology, ophthalmology, otolaryngology, and dentistry (−17.1%, −13.0%, −13.0%, −26.4%, −39.0%, and −22.7%, respectively) and in January 2021 for internal medicine, paediatrics, and surgery (−21.7%, −45.1%, and −23.8%, respectively).

Paediatrics experienced the largest percentage decrease in May 2020 and January 2021, followed by otolaryngology in April 2020 (Figure 2). After March 2020, the number of monthly outpatient claims in paediatrics, surgery, and otolaryngology decreased compared with that in the corresponding months in 2019. In addition, during the pandemic, the number of monthly internal medicine claims decreased, except in August 2020, and the number of monthly ophthalmology claims decreased, except in October 2020 and 2021. In contrast, the number of monthly outpatient claims for obstetrics and gynaecology in 2021 increased from that in the corresponding months in 2019.

The graphs summarising the percentage change in the number of outpatient claims, cost per day, and days per claim by speciality between 2020 and 2021 and the corresponding periods in 2019 are shown in Figure 3. Paediatrics and otolaryngology, in which the number of claims continuously decreased from the start of the COVID-19 pandemic, showed a large increase in the cost per day and a decrease in the days per claim. In addition, in internal medicine, ophthalmology, and dentistry, the cost per day increased by more than 10% in 2021 compared with that in the corresponding months in 2019 (Appendix A). In surgery, orthopaedics, and dentistry, the days per claim decreased by more than 5% in 2020–2021 compared with that in the corresponding months in 2019 (Appendix A).

## 4. Discussion

This study aimed to clarify the characteristics of the long-term trends in outpatient attendance by speciality in Japanese clinics between 2009 and 2021 and the short-term trends between 2020 and 2021. The changes in these trends during the COVID-19 pandemic were investigated. The following specialities showed statistically significant decreases in long-term trends during the COVID-19 pandemic, despite increased or stable outpatient numbers before the pandemic: internal medicine, paediatrics, ophthalmology, and otolaryngology. From March 2020 to 2021, the number of monthly outpatient claims in paediatrics, surgery, and otolaryngology decreased in all months compared with those in the corresponding months in 2019. Specialities that experienced a large decrease in the number of claims during the pandemic exhibited an increase in the cost per day and a decrease in the number of days per claim.

Between 2009 and 2021, internal medicine, paediatrics, ophthalmology, and otolaryngology showed statistically significant decreases during the COVID-19 pandemic, despite increased or stable outpatient numbers before the pandemic. The common feature of these specialities is that they often treat mucosa, such as the mouth and eyes. These procedures generate droplets and aerosols, which are the routes of COVID-19 infection [25,26,27]. Furthermore, in internal medicine and paediatrics, the establishment of a fever clinic for COVID-19 may have led to a decrease in visits by patients with other symptoms. Ophthalmology clinics had to cancel or postpone outpatient surgical procedures because of the pandemic, and postoperative follow-up visits decreased [28]. Many otolaryngology clinics refrained from performing procedures that involved the use of nebulisers in the early stages of the pandemic because of awareness of the risk of airborne transmission [29]. Annually from February through April, the number of patient visits in Japan usually increases because of cedar pollinosis; however, a decline in visits was observed during the pandemic [30].

For all specialities, the greatest decrease in outpatient visits during the pandemic occurred in April 2020 or January 2021, which were the months in which an emergency was declared by the government. Japan issued three emergency declarations for COVID-19: April–May 2020, January–March 2021, and April–September 2021. During emergency declarations, the government asked residents to stay at home to prevent the spread of infection [31]. The lack of personal protective equipment, especially face masks, in the early stages of the pandemic was a serious problem worldwide [32].

Between 2020 and 2021, the percentage of change in the number of outpatient claims compared with that in the corresponding months in 2019 by speciality showed a large and continuing decline in paediatrics and otolaryngology, which are the specialities most used by children. As infection control in children was challenging, parents may have chosen to refrain from taking their children to clinics. Previous studies have reported a greater decrease in visits by children compared with those by adults [18,33]. Moreover, in Japan, COVID-19 vaccination administered for the general population started in the summer of 2021; however, vaccination programmes for children (5–11 years) and infants (6 months to 4 years) began in February 2022 and October 2022, respectively [34]. A previous study reported that immunisation through vaccination programmes loosened individuals’ behaviour towards infection control [35,36].

Specialities with larger decreases in the number of outpatient claims tended to have higher costs per day, which could have been due to doctors performing more procedures or clinics giving higher-priced treatments in a single visit. In addition, a provisional insurance fee revision of 50 Japanese Yen (JPY) per visit for infection control practices from April to September 2021 could have contributed to this increase. For clinic visits by children, additional fees (December 2020–September 2021: 1000 JPY per visit for specialities other than dentistry, 550 JPY per visit for dentistry; October–December 2021: 500 JPY per visit for specialities other than dentistry, 280 JPY per visit for dentistry) were covered by provisional insurance fee revision. However, clinics may have been overtreating, such as by adding unnecessary treatments, because of reduced revenue as a result of the pandemic. This is consistent with the results of previous dental studies, which showed that private practitioners may compensate for their lack of income by providing more services than necessary when the number of patients is low [37,38].

This study had some limitations. First, the number of outpatient claims was analysed on a monthly basis, and the number of visits per patient within a month could not be ascertained. Second, due to the observational design, certain potential confounding factors may not have been fully addressed. Concerning changes in age structure, we tackled this issue by employing age-standardised proportions, although only including three age categories in our data may have been inadequate to control the effect of age. Changes in the number of healthcare facilities and insurance schemes could have influences the results. Still, no significant system changes in Japan affected the number of outpatients attendance during the study period. Third, for clinics with multiple specialities, all visits were classified into one ‘main’ speciality. Therefore, the number of claims for each speciality in the database we used does not exactly correspond to the actual number of claims issued for that speciality. However, since the ‘main’ speciality of the clinic is generally the one with the highest number of patients, it is assumed that most of the claims are for that speciality. In addition, since there is no evidence indicating a major change in the trend for clinics with multiple specialties over the study period, the impact on the analysis of long-term and short-term trends is considered limited [39]. Finally, our data did not include self-funded or publicly funded care. However, given Japan’s universal health insurance system, excluding self-funded care had a minimal impact on the objectives of this study. Additionally, as of December 2021, the proportion of publicly funded claims relative to all claims was only 1.86%, indicating that the exclusion of this subgroup had a limited effect.

## 5. Conclusions

Considering the long-term trends between 2009 and 2021, specialities that experienced a decrease in outpatient attendance during the COVID-19 pandemic were internal medicine, paediatrics, surgery, ophthalmology, and otolaryngology. For some specialities, the pandemic had a substantial impact, even in the context of long-term trends, considering the short-term trends between 2020 and 2021 in outpatient attendance to paediatrics, surgery, and otolaryngology specialities decreased. Specialities that experienced a large decrease in outpatient attendance during the pandemic experienced an increase in costs per day and a decrease in the days per claim. Specialty-specific preparedness requires consideration at the planning phase of healthcare policy during future public health emergencies. Further research is needed to understand the characteristics of specialties during public health emergencies.

## Figures and Tables

**Figure 1 ijerph-20-07133-f001:**
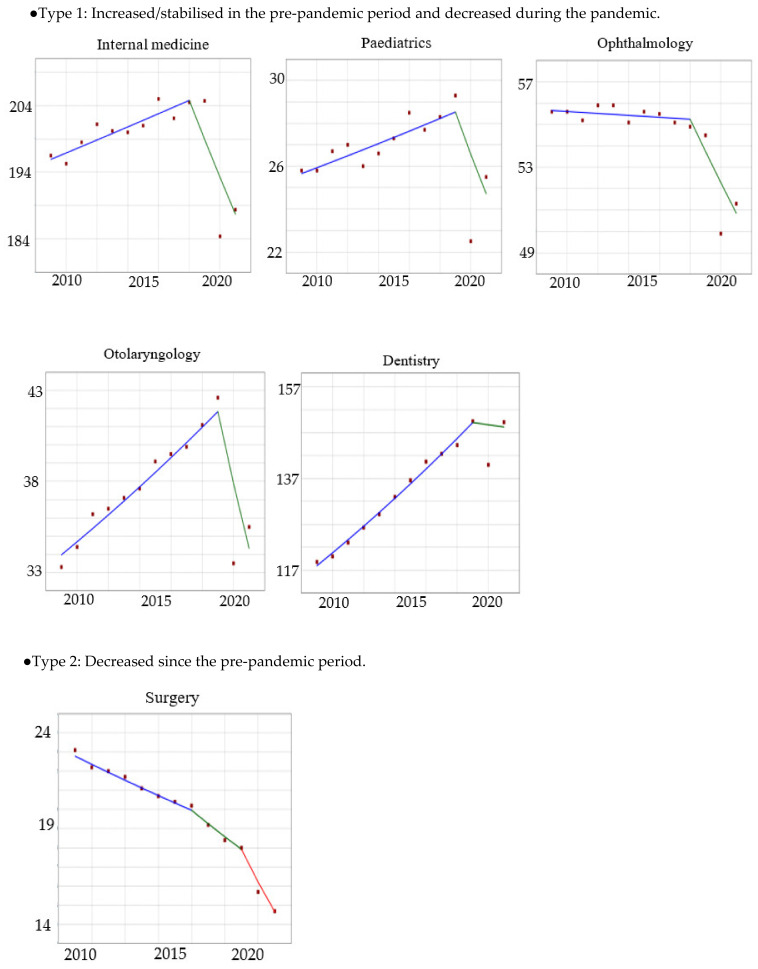
Type variety of the joinpoint regression on the age-standardised proportion of outpatient attendance by speciality between 2009 and 2021. Lines show continuous linear segments. X axis: year; Y axis: annually summarised age-standardised proportions. Segment 1 is represented as a blue line, segment 2 as a green line, and segment 3 as a red line.

**Figure 2 ijerph-20-07133-f002:**
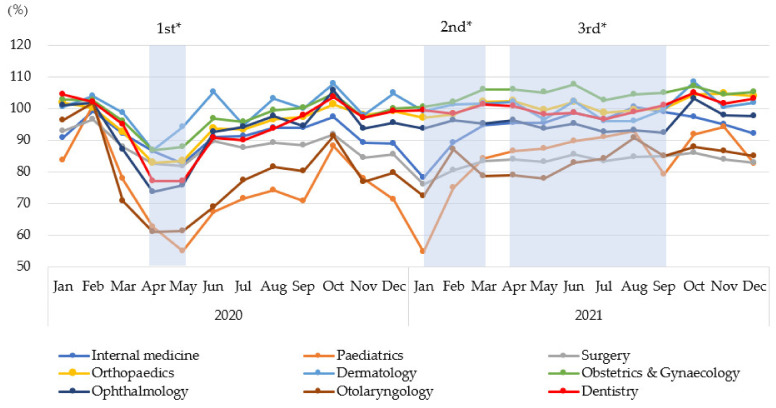
Percentage of change in the number of monthly claims by speciality between 2020 and 2021 and the corresponding periods in 2019. The percentage of change was centred at 100 for ease of interpretation. * The emergency declaration periods: 1st (7 April to 25 May 2020), 2nd (8 January to 21 March 2021), and 3rd (25 April to 30 September 2021).

**Figure 3 ijerph-20-07133-f003:**
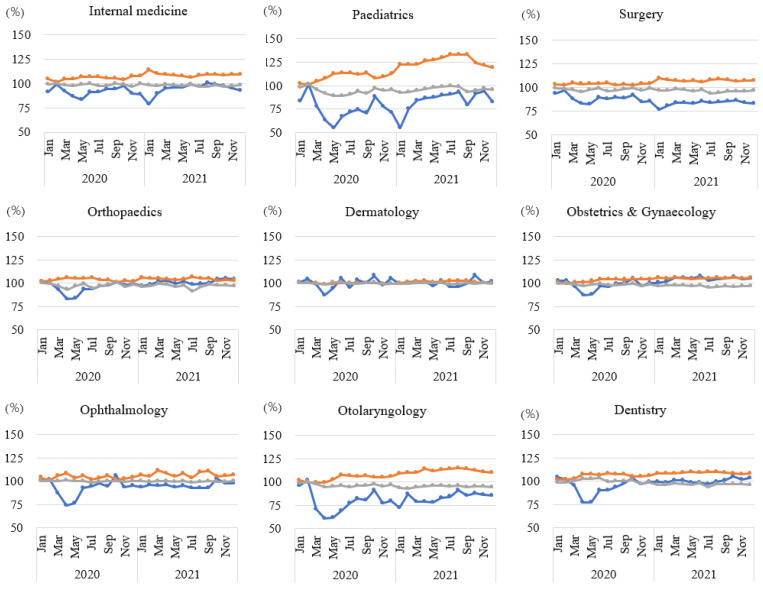
Trends in the number of outpatient claims, cost per day, and days per claim by speciality between 2020 and 2021 and the corresponding periods in 2019. Blue line: number of claims; orange line: cost per day; grey line: days per claim. The percentage change was centred at 100 for ease of interpretation.

**Table 1 ijerph-20-07133-t001:** Results of the joinpoint regression analysis on the age-standardised proportion of outpatient attendance by speciality between 2009 and 2021.

Speciality	Segment 1			Segment 2			Segment 3				
	Period	APC	95% Cl	Period	APC	95% CI	Period	APC	95% CI	AAPC	95% CI
Internal medicine	2009–2018	0.49 *	0.17, 0.91	2018–2021	−2.86 *	−4.78, −1.51	―	―	―	−0.36 *	−0.61, −0.13
Paediatrics	2009–2019	1.07 *	0.55, 2.72	2019–2021	−6.91 *	−12.93, −0.85	―	―	―	−0.31	−1.36, 0.72
Surgery	2009–2016	−1.87 *	−2.08, −1.47	2016–2019	−3.52 *	−4.09, −2.60	2019−2021	−9.53 *	−10.65, −8.60	−3.60 *	−3.78, −3.47
Orthopaedics	2009–2017	2.02 *	1.72, 2.63	2017–2021	0.17	−1.59, 0.99	―	―	―	1.40 *	1.06, 1.68
Dermatology	2009–2019	2.29 *	2.16, 2.56	2019–2021	0.29	−0.65, 1.41	―	―	―	1.95 *	1.81, 2.12
Obstetrics & Gynaecology	2009–2017	0.66 *	0.21, 0.86	2017–2021	2.29 *	1.27, 4.26	―	―	―	1.20 *	0.92, 1.34
Ophthalmology	2009–2018	−0.08	−0.39, 0.44	2018–2021	−2.72 *	−4.99, −1.37	―	―	―	−0.75 *	−1.07, −0.48
Otolaryngology	2009–2019	2.10 *	1.49, 3.08	2019–2021	−9.42 *	−14.72, −3.34	―	―	―	0.09	−0.87, 0.74
Dentistry	2009–2019	2.38 *	2.20, 2.75	2019–2021	−0.33	−2.33, 1.62	―	―	―	1.92 *	1.58, 2.16

Segments: continuous linear segments; APC: annual percentage change; AAPC: average annual percentage change; 95% CI: 95% confidence interval; * *p* < 0.05.

**Table 2 ijerph-20-07133-t002:** Number of monthly claims per 1000 people by speciality and percentage of differences between 2020 and 2021 and the corresponding months in 2019.

		Jan	Feb	Mar	Apr	May	Jun	Jul	Aug	Sep	Oct	Nov	Dec
Speciality	Year	N (% Change)											
Internal medicine	2019	235.2	211.2	216.0	210.3	201.2	200.9	207.9	195.8	197.8	207.3	214.0	224.2
	2020	214.0	209.7	199.0	182.1	166.7	183.3	190.0	184.0	186.1	202.0	191.3	199.5
		(−9.0)	(−0.7)	(−7.9)	(−13.4)	(−17.2)	(−8.8)	(−8.6)	(−6.0)	(−5.9)	(−2.6)	(−10.6)	(−11.0)
	2021	184.3	188.5	205.1	201.1	192.3	198.4	201.0	197.2	195.9	202.1	203.2	206.9
		(−21.7)	(−10.7)	(−5.1)	(−4.4)	(−4.5)	(−1.2)	(−3.3)	(0.7)	(−1.0)	(−2.5)	(−5.0)	(−7.7)
Paediatrics	2019	31.6	27.9	28.9	27.1	25.5	26.6	28.3	23.2	26.1	27.8	28.5	31.2
	2020	26.5	28.2	22.5	17.0	14.0	18.0	20.3	17.2	18.5	24.5	22.2	22.3
		(−16.1)	(1.2)	(−22.2)	(−37.2)	(−45.0)	(−32.5)	(−28.3)	(−25.8)	(−29.1)	(−11.7)	(−22.1)	(−28.6)
	2021	17.3	21.0	24.3	23.5	22.3	23.9	25.8	21.6	20.8	25.5	26.8	25.8
		(−45.1)	(−24.9)	(−15.7)	(−13.5)	(−12.6)	(−10.2)	(−9.0)	(−7.0)	(−20.6)	(−8.0)	(−5.8)	(−17.3)
Surgery	2019	19.4	18.4	19.2	18.6	18.0	18.4	18.8	17.8	18.0	18.5	18.7	19.1
	2020	18.0	17.8	16.9	15.4	14.8	16.5	16.4	15.9	15.9	16.9	15.8	16.4
		(−7.0)	(−3.3)	(−11.9)	(−17.2)	(−18.0)	(−10.3)	(−12.4)	(−10.6)	(−11.6)	(−8.2)	(−15.3)	(−14.4)
	2021	14.8	14.9	16.0	15.7	15.0	15.7	15.7	15.1	15.3	15.9	15.7	15.9
		(−23.8)	(−19.3)	(−16.5)	(−16.0)	(−16.8)	(−14.4)	(−16.5)	(−15.1)	(−14.8)	(−13.9)	(−15.9)	(−17.1)
Orthopaedics	2019	48.8	49.2	51.6	51.2	51.7	53.1	53.8	50.7	51.8	52.0	51.4	51.6
	2020	49.7	49.4	47.9	42.5	43.2	49.7	50.3	49.0	50.6	52.7	50.4	51.2
		(1.9)	(0.4)	(−7.3)	(−17.1)	(−16.5)	(−6.3)	(−6.4)	(−3.3)	(−2.4)	(1.4)	(−2.0)	(−0.6)
	2021	47.4	48.4	52.7	52.5	51.5	54.3	53.1	50.4	51.8	54.3	53.9	53.7
		(−2.9)	(−1.7)	(2.1)	(2.4)	(−0.3)	(2.3)	(−1.2)	(−0.5)	(0.0)	(4.4)	(4.9)	(4.1)
Dermatology	2019	38.0	38.7	43.8	42.6	44.0	45.7	48.6	45.5	43.2	41.0	41.5	41.9
	2020	38.2	40.3	43.3	37.1	41.5	48.1	46.2	47.0	43.3	44.3	40.7	43.9
		(0.6)	(4.1)	(−1.3)	(−13.0)	(−5.6)	(5.3)	(−5.1)	(3.2)	(0.1)	(8.1)	(−2.1)	(5.0)
	2021	37.7	39.2	44.6	43.6	42.5	46.8	46.8	43.7	43.1	44.5	41.8	42.7
		(−0.8)	(1.3)	(1.7)	(2.3)	(−3.4)	(2.3)	(−3.8)	(−4.0)	(−0.2)	(8.4)	(0.6)	(2.0)
Obstetrics and Gynaecology	2019	12.9	13.0	13.8	13.1	13.0	13.7	14.4	13.5	13.9	14.1	14.0	14.1
	2020	13.2	13.4	13.3	11.4	11.4	13.3	13.8	13.4	13.9	14.8	13.6	14.1
		(2.9)	(2.8)	(−3.9)	(−13.0)	(−12.1)	(−3.1)	(−4.1)	(−0.5)	(0.3)	(4.9)	(−2.7)	(0.0)
	2021	12.9	13.3	14.7	13.9	13.6	14.8	14.8	14.1	14.6	15.1	14.6	14.9
		(0.7)	(2.2)	(6.1)	(6.1)	(5.2)	(7.8)	(2.7)	(4.6)	(5.1)	(7.2)	(4.6)	(5.3)
Ophthalmology	2019	48.8	51.9	61.3	56.7	57.3	59.7	59.7	56.4	56.5	52.3	53.2	56.2
	2020	49.4	52.9	53.5	41.8	43.5	55.4	56.3	55.1	53.5	55.3	49.9	53.8
		(1.2)	(2.0)	(−12.7)	(−26.4)	(−24.1)	(−7.2)	(−5.7)	(−2.3)	(−5.3)	(5.8)	(−6.2)	(−4.3)
	2021	45.8	50.0	58.4	54.6	53.7	56.9	55.3	52.5	52.3	53.9	52.1	54.9
		(−6.1)	(−3.7)	(−4.6)	(−3.7)	(−6.2)	(−4.7)	(−7.4)	(−6.9)	(−7.5)	(3.1)	(−2.1)	(−2.3)
Otolaryngology	2019	38.6	48.5	62.0	47.4	40.5	39.1	37.2	32.1	35.1	39.5	42.2	40.6
	2020	37.2	49.4	43.9	28.9	24.8	26.9	28.8	26.2	28.2	36.2	32.5	32.4
		(−3.6)	(1.9)	(−29.1)	(−39.0)	(−38.7)	(−31.0)	(−22.5)	(−18.3)	(−19.6)	(−8.5)	(−23.1)	(−20.3)
	2021	28.0	42.2	48.8	37.4	31.6	32.4	31.3	29.1	29.9	34.8	36.5	34.6
		(−27.5)	(−12.9)	(−21.2)	(−21.0)	(−22.1)	(−17.0)	(−15.8)	(−9.2)	(−14.9)	(−11.9)	(−13.5)	(−14.9)
Dentistry	2019	139.4	144.4	158.5	149.5	146.4	156.2	157.8	147.3	146.7	148.8	151.1	156.2
	2020	145.8	147.6	150.6	115.5	113.1	141.8	142.1	138.1	143.7	154.6	146.9	155.2
		(4.6)	(2.2)	(−5.0)	(−22.7)	(−22.7)	(−9.2)	(−9.9)	(−6.2)	(−2.0)	(3.9)	(−2.8)	(−0.7)
	2021	138.8	142.4	160.6	150.9	143.9	154.4	152.7	145.9	148.5	156.2	153.7	161.4
		(−0.4)	(−1.4)	(1.4)	(0.9)	(−1.7)	(−1.1)	(−3.2)	(−0.9)	(1.2)	(5.0)	(1.7)	(3.3)

## Data Availability

Data are available from the Ministry of Health, Labour and Welfare website, under “Estimated Medical Expenses Database by MHLW” (https://www.mhlw.go.jp/bunya/iryouhoken/iryouhoken03/01.html; in Japanese).

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
