# Peer review of "Long- and Short-Term Trends in Outpatient Attendance by Speciality in Japan: A Joinpoint Regression Analysis in the Context of the COVID-19 Pandemic"

_ijerph, 2023, doi:10.3390/ijerph20237133_

Round 1

Reviewer 1 Report

Comments and Suggestions for Authors

Dear Authors, thank you for submitting this article and offering me the opportunity to review it. It was very interesting and I thought it was great work! Of course I recommend accepting it, but I would like to make some suggestions about analyses and visualizations that you could make to give it more impact (all entirely optional!) and also a few clarifications in the methods that are important. Finally I would like to propose an alternative joinpoint analysis approach that I think would be more informative. My points 1 and 3 are optional, but the clarifications requested in part 2 are important for clarity of the text. I hope you will consider all my suggestions, and thanks for submitting an interesting and well-written article.

Entirely optional proposals for improving presentation

The figures are all great, but I think for non-Japanese readers it would be helpful to present at least one figure where the onset of the pandemic and the states of emergency are shown, in order to make the potential relationship between these states of emergency and changes in attendance more clear.

Also, while the percentage changes obviously make sense, I think it would help to show the absolute numbers of claims, either in the paper or in a supplementary file.  When % changes are presented it's hard for the reader to clearly understand what they're looking at, and something closer to the raw data is useful.

Clarifications I think are important

I think I broadly understand the method, but there are a few details that I think would be helpful to be clarified. I list them here.

1. Age-standardization and data access: On line 88 - 89 you say you access data on children 0-5 years old, 6-69 year olds and over 70 year olds.  But then in 104-106 you say you calculated age-standardized proportions by age-class. Did you use the 6-69 year old data as a single age in this process? Or did you separately download claims by e.g. 5 year age group and standardize using these? I think if you did the former you should mention as a limitation that ageing might still affect your results (though this might not be important if you accept my proposal below for joinpoint analysis). 

2. Was your joinpoint analysis done using annual data? I feel like it was.

3. Can joinpoint handle the kind of proportions you presented here? Also in table 2 you say that the joinpoint analysis was for "age-adjusted prevalence". Is this a proportion? It seems it shouldn't be. I think you need to make clearer what the outcomes were in each analysis, and be careful about when you use the words "proportion", "claim", "prevalence" and so on. Please confirm these details.

3. A question about the suitability of your joinpoint analysis

It appears (see my clarification above) that you have run your joinpoint analysis on annual data. But I think the trend from 2009 - 2021 in Japan is subject to a lot of other issues that might interfere with modeling, e.g Great East Japan Earthquake, recovery from financial crisis, ageing, changes in national policies, etc. Since the purpose of this paper is to understand the effect of the covid pandemic, I wonder if it would be better to do the joinpoint analysis from e.g. 2018 - 2021, with monthly data? This would also fix the problem of having only two time points for trend assessment. A good example of this is pediatrics, which you report as "type 5: scattered" when it is really really really obvious that it saw a huge pandemic-related decline. I think if you start your analysis in 2018 by month you will find completely different results. Eg. dentistry would clearly have three trends, a pre-pandemic, during-pandemic, and post-pandemic trend that cannot be estimated in annual data but can be identified in monthly data.

I know it's super annoying when a reviewer suggests doing a different analysis, but I hope you will consider shortening the time frame for joinpoint analysis and using monthly data.

Reviewer 2 Report

Comments and Suggestions for Authors

This manuscript examines the short- and long-term trends of outpatient attendance by specialty in Japan, with a particular focus on the impact of the COVID-19 pandemic. The study utilizes data from Japan’s Estimated Medical Expenses Database between 2009 and 2021, provided by Japan’s Ministry of Health, Labour and Welfare (MHLW).

The number of monthly outpatient claims in clinics is used as an indicator of outpatient attendance for specialties such as internal medicine, pediatrics, surgery, orthopedics, dermatology, obstetrics and gynecology, ophthalmology, otolaryngology, and dentistry.

The study reveals that outpatient attendance in pediatrics, surgery, and otolaryngology decreased during the pandemic period compared to 2019. Moreover, internal medicine, ophthalmology, and otolaryngology experienced statistically significant decreases in outpatient attendance over the long term.

The study emphasizes the importance of specialty-specific preparedness to ensure essential outpatient services during future public health emergencies. 

1.The study design subsection needs to be further explained.(Line 74) .Please add why a retrospective observational study was conducted for this research. And please explain the exact statistical methods is used in this manuscript

2. If a clinic had multiple specialties, all visits were classified into one 'main' specialty, which could have led to the overestimation of major specialties. This could affect the accuracy of the analysis and the interpretation of the results.

3. The data used in the study did not include self-funded or publicly funded treatments. This exclusion may result in a biased representation of outpatient attendance in Japan.

4. The study is an observational study, so it does not discuss the potential confounding factors that may have influenced the observed trends in outpatient attendance. Without considering these factors, it is not convincing to attribute the changes solely to the impact of the COVID-19 pandemic.

5. The conclusion part may be further explained in order to suggest for potential healthcare policies change. 
